# Atomic Force Microscopy for the Study of Cell Mechanics in Pharmaceutics

**DOI:** 10.3390/pharmaceutics16060733

**Published:** 2024-05-29

**Authors:** Henrik Siboni, Ivana Ruseska, Andreas Zimmer

**Affiliations:** 1Pharmaceutical Technology & Biopharmacy, Institute of Pharmaceutical Sciences, University of Graz, 8010 Graz, Austria; henrik.siboni@uni-graz.at (H.S.); ivana.ruseska@uni-graz.at (I.R.); 2Single Molecule Chemistry, Institute of Chemistry, University of Graz, 8010 Graz, Austria

**Keywords:** AFM, cell mechanics, drugs

## Abstract

Cell mechanics is gaining attraction in drug screening, but the applicable methods have not yet become part of the standardized norm. This review presents the current state of the art for atomic force microscopy, which is the most widely available method. The field is first motivated as a new way of tracking pharmaceutical effects, followed by a basic introduction targeted at pharmacists on how to measure cellular stiffness. The review then moves on to the current state of the knowledge in terms of experimental results and supplementary methods such as fluorescence microscopy that can give relevant additional information. Finally, rheological approaches as well as the theoretical interpretations are presented before ending on additional methods and outlooks.

## 1. Motivation

The overseen significance of cell mechanics in pharmaceutics comes in part from the fact that pathological cells can change their stiffness (most famously cancer cells) [1,2,3,4,5], meaning that efficacy can be measured through cell mechanics and that drug delivery systems can be designed to target a characteristic mechanical property [6,7,8,9]. Even the mechanical environment of a cell can affect drug efficacy [10]. Even without any pharmaceutically active ingredient, a drug delivery system can still change the efficacy when entering the cell [10], thus making cell mechanics a novel way of tracking cellular uptake.

One of the most popular methods for testing cell mechanics is atomic force microscopy (AFM), which is already used for studying cancer cells [11] and evaluating anticancer drugs [12]. AFM has also been used to more generally follow drugs on the cell membrane [13]. The mechanics of the cell membrane have even been linked to uptake via endocytosis [14,15,16].

Despite this, research activity remains low, and only a couple of papers, if any, are published on cell mechanics in pharmacy each year, while cell mechanics overall has hundreds of papers published per year, according to the Web of Science (Figure 1).

This review identifies what a researcher will need to enter the field and lays out the research area’s present and future. For this, relevant studies are included, even if they are not nominally performed in the field of pharmacy, as well as more advanced techniques, particularly cell rheology. Due to the versatility and availability of AFM, focus is exclusively given to AFM-based techniques.

## 2. Basic Theory and Experimental Setup

For basic experiments, an AFM capable of so-called force–distance measurements in a cell culture medium is required. In the following, the necessary theoretical understanding and experimental equipment are presented in more detail.

### 2.1. Force Measurements with AFM

AFM is mostly used for actual microscopy in the form of topographical imaging with the most common mode being tapping mode, also referred to as intermittent contact mode [17], and this has already found applications in pharmaceutics [18].

For measuring the stiffness of a sample, the AFM tip is moved toward the cell at a speed $\vec{v}$ until it reaches the cell and deforms it at an indentation depth δ up to a set threshold force $F_{thres}$. However, the experimenter can only directly measure the deflection x of the cantilever and the distance $Z_{p}$ traveled by the piezo (Figure 2). The cantilever can be modeled as a spring for small deflections, for which the spring constant has previously been measured [19].

The distance d traveled by the tip can, therefore, be calculated as the piezo position minus the deflection of the cantilever:(1)$$d=Z_{p}-x$$

When the spring constant k of the cantilever is known, the force can be found by multiplying the spring constant with the following deflection:(2)F=kx

This gives the characteristic force–distance curves. Past the point of contact $d_{C}$, the tip is indenting into the cell, and the indentation depth δ is therefore as follows [19]:(3)$$\delta =d-d_{C}$$

Though several algorithms for determining this contact point exist, one should still make sure that this has indeed happened correctly [20].

As for the applied force, one should choose a threshold force $F_{\mathrm{thres}}$ that causes the desired deformation without destroying the cell sample. This can be tested in a pre-experiment. The significance of different deformation regimes is discussed in Section 2.2.1.

One should further take the drag force into account. Since cell mechanics measurements by necessity are performed in cell medium, the cell medium will always add a contribution $F_{drag}$ on top of the intended deformation force [21]. The higher the approach speed, the more important this contribution becomes, as the drag force is proportional to the speed of the probe tip:(4)$$F_{\mathrm{drag}}=\mu v_{\mathrm{tip}}$$
where μ is the viscous drag coefficient. This coefficient can be determined by moving the AFM probe through the medium without making contact with the sample [22].

For the interested reader, force-distance measurements can also be used to measure cell properties other than mechanics. A very popular avenue is single-molecule force spectroscopy in which the AFM probe is functionalized with molecules, resulting in just a few and ideally only one molecule at the very tip. When the tip approaches the surface, this molecule can bind to the surface. The corresponding unbinding force is observed in the force–distance curve when retracting the tip [23,24,25,26]. Force-distance curve can also be used to measure quantities such as dielectric constants [27]. However, these uses are outside the scope of this review.

### 2.2. Contact Mechanics

To estimate cell mechanics parameters, the in-contact part of the force–indentation curve must be analyzed by fitting a theoretical model to it. The simplest model is a straight line where the slope is the spring constant of the sample $k_{\mathrm{cell}}$. However, one of the most common fitting parameters is Young’s modulus E. This is similar to the spring constant, but it is independent of the size of the contact area and the size of the indenter. This is presented here.

#### 2.2.1. Stiffness and Young’s Modulus

At the very simplest, the cell can be modeled as a one-dimensional spring just like the AFM cantilever. If the cantilever is able to indent the cell by a distance δ by applying a force F, the spring constant of the cell, $k_{\mathrm{cell}}$, can be expressed by Hooke’s law:(5)$$k_{\mathrm{cell}}=\frac{F}{\delta }$$

However, this approach is inaccurate, especially if the indenting tip is much smaller than the cell. The solution is Young’s modulus which generalizes the spring constant and is defined by the relation between strain ε and stress σ (Equation (6)) [28]:(6)σ=Eε

One can also define an inverted Young’s modulus called elastic compliance J, in which case Equation (6) becomes [29]
(7)ε=Jσ

Strain is defined here as the compressed distance δ divided by the initial height $L_{0}$
(8)$$\varepsilon =\frac{\delta }{L_{0}},$$
while stress is the force applied per area A.
(9)$$\sigma =\frac{F}{A}$$

Just like the spring constant, the higher the Young’s modulus, the larger the stress needed for the material to undergo the same strain.

Plotting stress versus strain should yield a straight line according to this generalized Hooke’s law, but in reality, it only does so for sufficiently low values of strain (Figure 3). Going beyond this elastic region, the measured Young’s modulus is no longer reliable, and damage to the cell can occur [2,30,31,32].

Even if the cell does not break upon indentation, cells are known to deviate significantly from ideal elastic behavior due to their subcellular structure. Therefore, it is recommended that indentations do not exceed 200 nm [30] where the cortex—the outer part of the actin cytoskeleton—dominates [33]. Furthermore, since the apparent stiffness of any sample will be affected by the underlying substrate if the indentation is large enough, the indentations should not exceed 20% of the cell thickness without the proper model corrections [2,30,31,32].

Another value needed to describe cells is the Poisson ratio, which is defined as the change in trans-axial strain $\varepsilon_{\mathrm{trans}}$ with respect to the change in axial strain $\varepsilon_{\mathrm{axial}}$ [29]:(10)$$\nu =-{\left(\frac{d\varepsilon_{\mathrm{trans}}}{d\varepsilon_{\mathrm{axial}}}\right)}_{\varepsilon_{\mathrm{axial}}=0}$$

When a material like rubber is compressed (increasing $\varepsilon_{\mathrm{trans}}$), it will widen (decreasing $\varepsilon_{\mathrm{axial}}$), meaning that the Poisson ratio is positive (Figure 4b). It can be shown that a material with constant volume will have a Poisson ratio of 0.5. On the contrary, a material that can be compressed without becoming wider has a Poisson ratio of 0.

For simplicity, the Poisson ratio of a cell can be assumed to be 0.5 [34]; however, methods do exist for determining the Poisson ratio, and they have shown deviating values [35].

#### 2.2.2. Force Volume Mapping

On the other hand, one of the significant benefits of using an AFM for cell mechanics is the spatial resolution. By performing force–distance curves in several points on a lattice, an image of the sample properties can be constructed. Typically, one would make an image of the stiffness, but any quantity extracted from a force-distance curve can be mapped [36,37].

#### 2.2.3. Cell Indentation Models

When the tip of an AFM probe is indenting upon a cell, however, the deformation can typically not be described as an even compression of a cylinder. One of the most common approaches is the Hertz model, which makes the simplifying assumption that the sample is an ideal elastic material with no dissipative effects. Often, the tip is modeled as a paraboloid, in which case force vs. indentation is given by (Equation (11)) [38]:(11)$$F\left(\delta \right)=\frac{4}{3}\frac{E}{1-\nu^{2}}\sqrt{R}\delta^{\frac{3}{2}}$$
where R is the tip radius, and the Poisson ratio ν shows up in the nominator. However, many more models for other tip geometries and sample models exist [39].

### 2.3. Choice of AFM Probe

From the presented background knowledge, three key parameters stand out as crucial when choosing an AFM probe with which to study cell mechanics: spring constant k, resonance frequency $f_{\mathrm{res}}$, and tip radius R.

Firstly, both cantilever deflection x and sample indentation δ must occur in order to generate a force–indentation curve. This is determined by the spring constant k, as a too-stiff cantilever will deform the cell but not show any force, while a too-soft cantilever will not deform the cell and will instead only reveal the force needed to bend the cantilever itself. For this reason, the cantilever stiffness should be roughly equal to the spring constant of the cell $k_{\mathrm{cell}}$:(12)$$k\sim k_{\mathrm{cell}}$$
where $k_{\mathrm{cell}}$ will lie in the range of $0.01-0.6\frac{nN}{nm}$, depending on the cell line [21].

Secondly, the resonance frequency $f_{\mathrm{res}}$ of the cantilever must be chosen so that no resonant movement will disturb the measurement. This is less of a risk if a full indentation curve is taken only once a second, but some AFMs can probe thousands of times per second. It, therefore, holds that the resonance frequency must be much larger than the intended probing frequency $f_{\mathrm{probe}}$ [21]:(13)$$f_{\mathrm{res}}\gg f_{\mathrm{probe}}$$

This has the further benefit that the cantilever is protected against low-frequency noise such as footsteps or building vibrations [19].

Note, however, that the resonance frequency in the cell medium $f_{\mathrm{med}}$ will not be the same as in air since the cell medium will introduce the aforementioned drag μ as well as add to the effective mass m of the probe. The latter is because the medium close to the cantilever will move with it. After modeling the cantilever as a dampened harmonic oscillator, the new frequency is given by [19]:(14)$$f_{\mathrm{med}}=\sqrt{\frac{k}{m}-\frac{1}{2}{\left(\frac{\mu }{m}\right)}^{2}}$$

It is, therefore, advisable to measure the actual resonance frequency, once the AFM probe is submerged in the medium.

While the resonance frequency increases with increasing spring constant, it is also determined by effective mass (as shown in Equation (14)) as well as the dimensions of the cantilever. AFM probes can, therefore, be constructed to have a low spring constant but with a high resonance frequency [19].

The most deciding part of choosing the right AFM probe, however, is the tip shape, which is quantified by its radius of curvature R. A very large radius can be achieved by using a microbead as a tip, in which case a large part of the cell is deformed, and simple models like Equation (11) work well to find Young’s modulus. The measured stiffness will then correspond to the cell’s overall stiffness. Since cells are inhomogeneous, it can be instructive to use a large radius indenter [34] to arrive at more reproducible results. Alternatively, a smaller radius can deliver more location-specific mechanical properties—a particular strength of AFM—and it is, therefore, well suited for force volume mapping. However, the models needed for smaller tips are more complex [21].

The studies included in this review have used tip radii down to 2 nm (e.g., [40]), over 20 nm (e.g., [41]), to 60 μm (e.g., [42]), and even a flat wedge (e.g., [43]) which effectively has an infinite radius of curvature.

### 2.4. Cell Considerations

To work with living cells, a well-functioning cell culture setup is a prerequisite. Before AFM measurements, the cells of interest need to be isolated and immobilized on a flat surface.

To separate single cells from a mass, one can use the following techniques: flow cytometry, manual cell picking, optical tweezers, random seeding, dilution, and so on [44]. For this purpose, the cells must be enzymatically treated (for example, using trypsin) in order to be separated from the surface to which they had been adhering, or from one another. This can lead to a change in the cell surface, given that trypsinization causes cell rounding. Furthermore, since trypsinization or the use of EDTA (2,2′,2″,2′″-(Ethane-1,2-diyldinitrilo)tetraacetic acid) degrades the extracellular matrix and disrupt protein–protein interactions between cells, this can lead to the loss of some membrane receptors on the cells [45,46]. The procedures leading up to cell separation and seeding can also result in membrane fragments being deposited on the surface, which can also influence the AFM measurement.

As for cell immobilization, there are various options, depending on the chosen cell line. For microscopy purposes, anchorage-dependent cells are most commonly seeded on glass-bottom dishes that are manufactured using thin, borosilicate glass. Nevertheless, there are some fastidious cell lines that require pre-treatment of the growth surface to attach to it and proliferate. Commonly, in this case, the surface is pre-treated using collagen, serum, fibronectin, gelatin, or poly-L-lysine. When it comes to suspension cell lines, their immobilization is usually carried out by pre-treating the glass surface with positively charged poly-L-lysine. The polymer interacts with the negatively charged cell membrane, followed by the attachment of the suspended cells to the surface. The mechanical properties of the chosen immobilization surface (extracellular matrix) are of great importance, given that its stiffness can affect various intracellular processes [47].

Temperature is another important factor for studying cellular physiological and pathological processes. Most animal cell lines require 37 °C for optimal growth. It is necessary to maintain the atmospheric temperature as close as possible to the physiological temperature, given that temperature variations can lead to changes in protein and enzyme activities, as well as membrane fluidity. Furthermore, exposure to low temperatures can reduce cellular viability, inhibit the cycle progression, and upregulate apoptosis [48,49,50].

### 2.5. Available Software

Software solutions exist to analyze force-distance measurements. While some are functional, but prohibitively expensive—such as *MountainsSPIP*—or only come commercially with the AFM, there is also the generally popular AFM software *Gwyddion* (by the time of publication, the latest version is 2.65) [51]. *Gwyddion* can analyze force–distance curves, but only for a limited number of models and file types. Python programs with Graphical User Interfaces such as *PyJibe* [52], *Nanite* [53] (supporting machine learning), and *PyFMLab* [54] (supporting rheology) have been published, but unfortunately not as stand-alone software, thus making them inaccessible to researchers without knowledge of Python.

To our knowledge, one of the best software and the best freeware currently available is the open-source *AtomicJ* [55], which can open a wide variety of file formats and offers several methods for the analysis of force-distance curves as well as analysis corrections. For someone new to the field, *Gwyddion* and *AtomicJ* are excellent for achieving first results.

However, it can also be difficult for new users to choose the correct settings and support the full range of experiments described in this review. The lack of software solutions was noted by Butt et al. in 2005 [19]. Some studies, therefore, report using their own programs written in programming languages or mathematical programs, such as MATLAB [56] and Igor [22].

## 3. Current Results on Cell Stiffness

With this basic knowledge in hand, an overview of the current empirical results is presented in this section.

### 3.1. Empirical Trends

To analyze the literature, articles were collected using the Web of Science as well as from previous reviews on nanoparticles [57] and chemotherapeutics [12]. Further papers were added as they were discovered or referenced, and the most interesting messages are presented in this review. Some of the papers [14,40,41,42,43,56,58,59,60,61,62,63] were classified in a meta-analysis (Figure 5 and Appendix A).

It was found that the studies, to a large extent, are not performed by pharmaceutical institutions, and they are also mostly not published in pharmaceutical journals, thus making the research area less exposed to a pharmaceutical audience. Most studies did not cover cell rheology, and a sizable fraction of studies use a non-biomimetic hard substrate like glass, thus showing clear areas where further work can be conducted.

#### 3.1.1. Cytotoxicity Assay

One of the most practical ideas is to use cell stiffness to study cytotoxicity. Given that the mechanical state of a cell is the result of ongoing intracellular effects, mechanical insights can give information about the general state of the cell [64]. Cytotoxicity is mostly associated with anticancer drugs, where the cytotoxic outcome is the expected one. Given that the invasiveness of cancer cells relies on extensive changes in the cytoskeleton, actin and microtubule rearrangement are processes that are constantly occurring. For this reason, most anticancer drugs have a targeted effect towards actin or microtubule polymerization. Depending on the type of drug, its mechanism of action, as well as the cancer cell line, the cytotoxicity can be expressed as increased roughness and decrease in size (as it is observed in the case of apoptosis-inducing drugs) [12]. When it comes to cellular elasticity, the observed outcome can vary based on the aforementioned factors. For instance, Lin et al. observed an interesting effect of paclitaxel on melanoma cells [65]. Paclitaxel works by blocking the mitotic cycle, leading to apoptosis. In the first hours of treatment, there was no change in elasticity, followed by decreased elasticity in the 30th hour of treatment. Paclitaxel was also used on prostate cancer by Ren et al., where an increase in cell stiffness was observed [66]. On the other hand, Raudenska et al. reported an increase in stiffness correlated with the cytotoxicity of cisplatin [67]. The authors suggest that the observed increase is due to actin accumulation and alteration of microtubule disassembly caused by cisplatin [68].

The effect that cytotoxic drugs have on the cell’s mechanical properties does not always have to be associated with changes in the cytoskeleton. Changes in cellular stiffness due to toxicity have been reported for drugs that affect the cell’s glycolytic activity. In this case, most of the anticancer drugs were associated with an increase in cellular stiffness [12].

This idea was first presented by Zimmer et al. in 2014, who argue that cell stiffness can be used to study cytotoxicity [42]. In their study, they showed that non-toxic SiO_2_ nanoparticles did not dramatically affect cell stiffness, while cytotoxic ZnO nanoparticles did alter cell stiffness. However, they also concluded that some cells increased in stiffness while others decreased in stiffness. They argued that the method’s strength comes from being able to measure cytotoxicity at the level of single cells. Murashko et al. support the idea that cytotoxicity can be measured but found that cytotoxicity overall decreases the stiffness of a population (Figure 6) [69].

They also found that this method is more reliable than morphological cell measurements for estimating cytotoxicity. Pastrana et al. argue that the softening is due to damage to the F-actin in the cytoskeleton, which is caused by radical oxygen species [58].

Independently, cytotoxic amyloids—known as neurodegenerative diseases—have been investigated in a number of studies [41,60,61].

#### 3.1.2. Uptake Tracing

A different idea is to follow the uptake of a drug through cell mechanics. The cellular surface represents an intricate and multifaceted system, given its complex structure and function. Composed of a lipid bilayer, extra-, intra-, and transmembrane proteins, as well as the actin cytoskeleton, it is involved in multiple processes such as differentiation, motility, adhesion to the extracellular matrix, and vesicle trafficking [70]. Therefore, it is only reasonable to expect that the cell surface mechanics have a great influence on cellular homeostasis.

Endocytosis is a process that originates from the cellular membrane and is involved in maintaining intracellular homeostasis by regulating the flow of nutrients and fluids inside the cells. What is more, endocytosis is the mechanism mostly described as used by viruses and nanoparticles as a mechanism of cellular internalization [71,72]. For membrane curvatures to form and endocytosis to occur, certain bending and membrane tension energy barriers need to be overcome. In the event of low membrane tension and the presence of extracellular cargo (i.e., nanoparticles), the generation of membrane curvatures is stimulated [16]. This process is followed by a reduction in cell surface area and an increase in cell volume. Finally, endocytosis results in increased membrane tension that acts inhibitory on any further uptake processes [73].

Under conditions of low membrane tension, the uptake process via vesicle formation occurs rather fast, with 20 min being reported as needed for the uptake of nanoparticles and viruses with a size of 25–30 nm [71]. However, under increased tension, the density of active endocytic pits decreases, and the actin cytoskeleton supplements the energy needed for vesicle formation and their subsequent internalization. Therefore, endocytic events dependent on actin polymerization will be slower than their counterparts [74]. The way one cell identifies and reacts to the changes in the tension is by the activity of mechanosensors (i.e., molecules and signaling pathways that are activated by changes in the extracellular matrix). The mechanism involved in activating the actin cytoskeleton during increased membrane tension includes the small GTPase Rho and its effector, Rho Kinase (ROCK) [75]. Furthermore, membrane-bending proteins (ones with transmembrane BAR domains) can also aid in endocytosis in cases of high membrane tension by producing forces to deform the membrane [76]. The striking difference in overall membrane tension and stiffness between healthy and diseased tissue, as well as the dysregulation of Rho/ROCK, especially observed in cancer, can be exploited as a means to achieve an increased uptake of nanoparticles in the diseased cells.

Joseph and Liu provide an elegant overview of the mechanical factors that regulate endocytosis, such as the membrane tension and mechanical properties of the extracellular matrix [16]. Furthermore, they also elaborate on the mechanical properties of the cargo and their influence on endocytosis. Interestingly enough, they report that stiffer cargo is more readily internalized compared to soft cargo.

AFM can be used as a method complementary to a liquid biopsy, for example, to first identify the mechanical phenotype of the cell, and then evaluate the potential for uptake of nanoparticles as drug delivery systems. Gold nanoparticles—a popular system for drug delivery [77]—have had their uptake traced over time within minutes by Kulkarni et al. in 2021 (Figure 7) using stiffness [15].

The cells experienced a temporary drop but returned to a stable value within 20 min which the authors attribute to the cell’s plasma membrane instead of the cytoskeleton. Notable, the 20 min time frame is comparable to the value reported by Gao [71] for 25–30 nm nanoparticles.

In 2022, Kulkarni et al. published a follow-up study, in which they were able to distinguish receptor-dependent and -independent endocytosis using the poroelastic model (Section 4.2.6) [14]. For this, they needed to measure force over time (Section 4).

### 3.2. Complementary Methods

As a complementary method to AFM, a natural choice is to combine force volume mapping with optical microscopy. A 2020 study transfected cells with polylactic-co-glycolic acid–polyethylene glycol nanofibers and used fluorescence microscopy to show that their position within the cell correlated with increased cell stiffness (Figure 8) [59]. Other microscopy techniques like immunofluorescence or transmission electron microscopy can likewise be used to monitor the drug or other changes inside the cells [58].

Regular optical microscopy is particularly useful for correctly placing the AFM probe above the cell [78], as well as monitoring the state of the cell [60].

A different approach is to track the chemical changes inside the cell to explain AFM results. A 2014 study by Fang et al. studied the effect of N-methyl-D-aspartate and found that they could indeed see a significant stiffening of the treated cells [40]. Using Western blotting, they correlated this with an increased concentration of RhoA—which activates the cell’s motor proteins—and myosin IIb, which is a motor protein. Topographical images and deformation images of the cells were likewise made with the same AFM as the stiffness measurements.

The existence of free oxidants has similarly been followed with laser-enabled analysis and processing to explain cytotoxicity [58]. If the interest is in cytotoxicity, an obvious choice is to also compare AFM results with established cytotoxicity assays such as lactate dehydrogenase and MTT assays [61] or flow cytometry [58].

### 3.3. Simplest Way to Get Started

Based on the preceding overview, the following simple workflow will fit a new experimentalist who wishes to measure the effect of a drug (Figure 9).

Having a pre-established cell culture setup, adherent cells should be seeded in a Petri dish to minimize cell movement during AFM measurements (Figure 9, (1)). The cells should then be treated with the drug of interest while a comparable population is left untreated as a control (Figure 9, (2,3)). In each Petri dish, as many cells as possible should be measured for optimal statistics and to account for different subpopulations. While mapping can be performed, it is time-consuming and it is, therefore, recommended to measure a smaller square in the middle of the cell [63] or simply in the center if a large colloidal tip is used (Figure 9, (4,5)) with an indentation depth of 200 nm [2]. For both treated and control cells, measurements should be performed before and after transfection and, ideally, as a time series with multiple points to determine the time scale of mechanical changes [13]. Finally, Young’s modulus can be extracted from the data using software like *Gwyddion* or *AtomicJ* (Figure 9, (6)).

If available, control of CO_2_ levels and temperature should also be implemented in the AFM stage. While more advanced strategies certainly exist, this workflow gives a solid starting point for arriving at the first results.

## 4. Rheology and Realistic Cell Modeling

This section expands upon the stiffness picture of the previous sections and sheds light on the less common, but valuable, rheological measurements and modeling of cells.

### 4.1. Rheology and Force-Time Curves

In the preceding text, it was assumed that a cell is an ideally elastic material, which means that measurements are time-independent. An experiment should, therefore, be performed at equilibrium conditions or in practice so slowly that this approximation is valid. However, cells are not ideally elastic but show viscous time-dependent behavior as well. There is, therefore, an active area of research investigating not only cell elasticity but the more complete cell rheology [79,80,81]. While such experiments (Figure 10) can be more complex than elasticity measurements, the field of rheology is also already well established in other research areas [82]; therefore, a researcher can reapply this knowledge for cell rheology.

One approach is to infer rheology from the regular force measurements as a function of time [79,83] (Figure 10a), and simply from the force-distance curve, one can calculate dissipated work due to the hysteresis between approach and retract [41] (Figure 2c). A different approach is to repeat the nanoindentation at different tip velocities [84]. However, for time-sensitive samples, such as cells, this may not be an option.

The more common angle to rheology, which will be the focus here, is to measure for a period of time between approach and retract while the tip is in contact with the sample [85]. During this waiting time, either the piezo position or the force is kept constant (static rheology), corresponding to the so-called stress relaxation and creep compliance, respectively, or one of the two is set to oscillate (dynamic rheology) (Figure 10). Like with force–distance curves, force volume images of the rheological properties can also be created [86].

#### 4.1.1. Static Rheology

Typically, for force relaxation, the approach happens faster than with a regular stiffness measurement. That way, the time for approaching can be neglected and the strain or stress can therefore be considered a simple step function, i.e., zero before and a constant value after the time of contact. In constant force mode, the feedback is turned on to keep the deflection of the cantilever constant. For a viscous sample, this necessitates that the piezo keeps moving forward into the sample. This means that the tip position as a function of time is the main observable. This is called creep compliance [85] and can be analyzed for AFM using the force clamp method [22].

For stress relaxation [87], the strain is kept constant (Figure 10c):(15)$$\dot{\varepsilon }=0$$
whereas for creep compliance, the stress is constant (Figure 10b):(16)$$\dot{\sigma }=0$$

The Hooke relation is modified so that Young’s modulus is now time-dependent
(17)$$\sigma \left(t\right)=E\left(t\right)\varepsilon_{0}$$
and called the relaxation modulus. A simple way of rewriting Equation (17) to fit a force–time curve from an AFM experiment is as follows [87]:(18)$$F\left(t\right)=E\left(t\right)\frac{C_{n}}{1-\nu^{2}}\delta_{0}^{n}$$
where $C_{n}$ and n are form factors and $\delta_{0}$ is the constant indentation. This equation is practical in that the relaxation modulus and the force are simply proportional, meaning that the characteristic time scale is unaffected.

Similarly, the strain at constant stress can be described using
(19)$$\varepsilon \left(t\right)=J\left(t\right)\sigma_{0}$$

Some studies use variations upon these two techniques, such as several steps during a single measurement [88,89,90].

#### 4.1.2. Dynamic Rheology

In the oscillatory mode (Figure 10d), the AFM piezo is oscillated at a controlled frequency ω and amplitude $\varepsilon_{0}$ [91]—possibly through a range of frequencies or amplitudes. This means that the strain applied to the cell is sinusoidal:(20)$$\varepsilon \left(t\right)=\varepsilon_{0}cos\left(\omega t\right)$$

Consequently, the strain will likewise oscillate with an amplitude $\sigma_{0}$ and with the same frequency:(21)$$\sigma \left(t\right)=\sigma_{0}cos\left(\omega t+\varphi \right)$$
where φ is a phase shift. For an ideally elastic material, strain and stress are proportional, indicating that φ is equal to zero.
(22)$$\sigma =E\varepsilon =E\varepsilon_{0}cos\left(\omega t\right)$$

On the other hand, a fully viscous material has stress proportional to the strain rate, where η is the proportionality factor known as the viscosity:(23)$$\sigma \left(t\right)=\eta \dot{\varepsilon }\left(t\right)=-\eta \varepsilon_{0}\omega sin\left(\omega t\right)=\eta \varepsilon_{0}\omega cos\left(\omega t+\frac{\pi }{2}\right)$$

In such a case, the phase shift is equal to 90°. To have a useful measurable quantity, the complex modulus E has therefore been defined. Rewriting strain and stress as complex exponentials
(24)$$\varepsilon \left(t\right)=\varepsilon_{0}e^{i\omega t}$$
(25)$$\sigma \left(t\right)=\sigma_{0}e^{i\left(\omega t+\varphi \right)}$$
the usual definition of Young’s modulus yields:(26)$$E=\frac{\sigma \left(t\right)}{\varepsilon \left(t\right)}=\frac{\sigma_{0}}{\varepsilon_{0}}e^{i\varphi }$$

This can also be written algebraically as
(27)E=E′+iE″
where E′ is the storage modulus—the elastic contribution—and E″ is called the loss modulus—the viscous contribution. As an alternative to E″, one can report the loss of tangent $tan\left(\varphi \right)$, which is 0 for an ideal elastic material and 1 for an ideal viscous material. In this way, the complex modulus can describe a viscoelastic material [29]. Note that some studies might instead refer to the shear modulus G [62], which is more typical in macroscale rheology and is found by torsional deformation. For the simple case of a small deformation of a homogeneous and isotropic material, Young’s modulus and the shear modulus are related by the following [28]:(28)$$G=\frac{E}{2\left(1+\nu \right)}$$

The shear modulus otherwise follows the same principles as the complex Young’s modulus described above [29].

### 4.2. Models of Cell Mechanics

Having presented the major approaches in cell mechanics—cell stiffness and cell rheology—it is time to consider in more detail how an experimental outcome may be interpreted. A major challenge in the field is the lack of a single theoretical model to causally interpret the raw results from cell mechanics experiments [2,30,57]. However, several models of varying degrees of complexity exist (Figure 11) [2,29].

#### 4.2.1. Ideally Elastic Cell

The most basic model is to consider the cell as being ideally elastic (Figure 11a). As has been described earlier in this paper, the cell is in that case represented by either a simple spring constant or by the related Young’s modulus.

#### 4.2.2. Spring-Dashpot Models

However, a spring is not sufficient, if rheological properties are to be taken into account. The most straightforward solution is to introduce a viscous dashpot which—like the spring—also relates stress and strain. However, for the dashpot, the stress is proportional to the rate of strain $\dot{\varepsilon }$ (Equation (29))
(29)$$\sigma =\eta \dot{\varepsilon }$$
where the proportionality factor η is the viscosity [29].

For a constant stress experiment, the strain of a dashpot is linear in time, thereby introducing the desired time-dependent behavior. It also facilitates hysteresis because the sign of the stress changes with the strain rate. By combining one or more springs (elastic contributions) and dashpots (viscous contributions), phenomenological models can be constructed to fit the rheological measurements (Figure 11b). They can be combined in series as a Maxwell material or in parallel as a Kelvin-Voight material. Further simple models include the Standard Linear Solid and the five-element Maxwell model (Figure 12) [29].

#### 4.2.3. Power Law Models

Moving away from spring-and-dashpot models, power law models have also turned out to work [62]. In such cases, the force–time curve is fitted to a power law where the exponent is called the fluidity. It has been shown that fluidity and stiffness are strongly correlated in the case of cells [93].

The creep compliance $J\left(t\right)$ can be expressed as follows [22]:(30)$$J\left(t\right)=J_{0}{\left(\frac{t}{t_{0}}\right)}^{\beta }$$
where $t_{0}$ is an arbitrary constant that can be set equal to 1 s, while $J_{0}$ is the creep compliance as time $t_{0}$. The power exponent or fluidity β is 0 for purely elastic samples and 1 for purely viscous samples, while viscoelastic samples are in between [2]. While the creep compliance is not directly equal to the indentation δ, they are at least proportional:(31)$$\delta \left(t\right)\propto J_{0}{\left(\frac{t}{t_{0}}\right)}^{\beta }$$

Via fitting, β can be determined. If the proportionality factor is known, $J_{0}$ can likewise be found and—from that—an apparent Young’s modulus [93]:(32)$$E_{\mathrm{app}}=\frac{1}{J_{0}}$$

Similarly, for oscillatory experiments, the stiffness can also follow a power law as a function of the frequency ω of indentation [91]:(33)$$E\left(\omega \right)=A\omega^{\alpha }$$
where A is a pre-factor and α is the power law exponent which is likewise 0 for purely elastic and 1 for purely viscous materials.

As a complication, different power laws might hold for different time scales. To some extent, spring-dashpot models can also produce such power laws [94].

Though purely phenomenological on their own, power laws can be used for comparing samples [2] and some of the following models do support power laws.

#### 4.2.4. Polymer Network

One more biological approach focuses exclusively on the cytoskeleton and therefore describes the cell as a polymer network (Figure 11c) with polymer stiffness (persistence length) and the density of crosslinking as model parameters. However, it turns out that this model is still not practically applicable to the actual cell’s cytoskeleton, and actual cells are orders of magnitude stiffer than an equivalent polymer network. It has been suggested that this is due to the fibers being constantly under stress; a fact which has been observed in experiments [2]. Despite the shortcomings, polymerization of actin was used by Gao et al. to explain why stiffness increased upon treatment with beta-amyloids, and the authors used immunofluorescence to test whether the cytoskeleton was indeed affected [61].

#### 4.2.5. Semi-Permeable Balloon

At the other end of the spectrum, the attempt exists to completely disregard the cytoskeleton and instead focus on the cell membrane (Figure 11e), which itself has a Young’s modulus $E_{m}$ and Poisson ratio $\nu_{m}$ [60]. When compressed, the cell membrane is stretched while the cell volume stays the same. Considering the cell as a spherical water balloon, the measured force arising from the stretching is given by
(34)$$F=2\pi \frac{E_{m}}{1-\nu_{m}}bR_{0}\varepsilon^{3}$$
where b is the membrane thickness and $R_{0}$ is the radius before compression. In this case, the membrane stiffness is the fitting parameter. This approach works best if a large AFM probe like a bead or wedge is used and if the cell is close to this ideal sphere.

In principle, models for describing unilamellar vesicles in terms of surface tension [95] can also be used.

If an influx of ions into the cell is suspected, the change in stiffness can then somewhat contradictorily be attributed to an increase in osmotic pressure Π according to
(35)ΔF=ΠS
where ΔF is the change in measured force and S is the contact area with the probe. The osmotic pressure is related to the change in concentration ΔC inside the cell by the ideal gas constant R and the temperature T:(36)$$\Delta C=\frac{\Pi }{RT}$$

Thereby, one arrives at a concentration that might be possible to measure independently. This model has been used to explain the effect of amyloids [60] and cytotoxicity in general [42].

#### 4.2.6. The Poroelasticic Cell

One relatively new approach to cell mechanics is poroelasticity, introduced in 2013 [96]. Here, the cell consists of a fully elastic framework—well applicable to the cytoskeleton, organelles, and macromolecules—containing a fully viscous fluid—corresponding to the cytosol—which is squeezed out when the cell is compressed. One can, therefore, consider the cell as a wet sponge (Figure 11d). By relating model parameters such as pore size to the actual cell architecture, the poroelastic model in principle makes real predictions possible [30,31,96]. The model has been shown to work well for the cytoplasm apart from long-term/low-frequency behavior [30,96]. Poroelasticity was used by Kulkarni et al. to show that receptor-dependent endocytosis of gold nanoparticles leads to changes in diffusion coefficient and pore size while receptor-independent endocytosis did not [14]. According to Moeendarbary et al., poroelastic results can mainly be explained by the actin cytoskeleton and its myosin motor proteins [96].

#### 4.2.7. Soft-Glassy Cell

A relatively special approach is to describe the cell as a soft-glassy material (Figure 11f) [30,89,97,98]. In soft-glassy rheology, the cell consists of many substituent particles with each particle trapped in a local potential minimum. When the cell is disturbed by applying a force, a flow of material occurs as the particles are pushed to lower minima. This fits well with the realistically crowded interior of a cell. The soft-glassy model gives rise to a weak power law, except for high frequencies.

Furthermore, it was noted already in 2001 that the power law exponent, referred to as x−1, could be used to quantify drug potency and that this exponent could directly predict rheology (Figure 13) [97].

#### 4.2.8. Finite Element Analysis

A more computational approach is to model the sample using Finite Element Analysis in which each subcellular component is modeled and the cell shape can be freely chosen [99]. These simulations have been compared to the traditional indentation models to determine their accuracy [100]. They can also be used to model how the actin cytoskeleton must be arranged to reproduce the experimental results [59].

#### 4.2.9. Models vs. Real Cells

Important to remember when comparing cells to traditional materials is that the material is often assumed to be passive (no stored non-mechanical energy) and non-aging (no mechanical change over time) [29]. This holds well for materials like rubber but is famously inaccurate for describing living cells which really fall into the category of active matter [2,101]. One study observed irregular force relaxation measurements and interpreted them as the cells actively responding, but most cells did not exhibit this active response [62]. However, if such active responses can be measured reliably, they could become a measure of viability and cytotoxicity.

Furthermore, it has been implicitly assumed that the cells are linearly viscoelastic—meaning that relaxation modulus and creep compliance are time-dependent, but not load-dependent—which is also not the case [30,102]. Research is likewise being conducted to take plasticity—non-reversible deformations (Figure 3)—into account [103].

A realistic cell generates its own forces via molecular motors and actin polymerization and can likewise respond to applied forces—the so-called mechanotransduction—via stress-activated ion channels, many of which are unknown. This can even be affected by the type of substrate, making the mechanical environment an important factor to control [30,104] and should ideally be close to in vivo conditions [2].

One can, however, compare measured values to known cellular properties, such as by comparing the characteristic time of rheology experiments to the time scale of the processes (Figure 14), to at least make semi-quantitative interpretations. For instance, Moreno-Flores et al. performed stress relaxation microscopy with the two-component generalized Maxwell model, thereby arriving at two different relaxation times [86]. They attributed the shorter time of about 0.1 s to the cell membrane while the longer time of ~1 s was attributed to the cytoskeleton.

Regardless, for drugs that are supposed to directly affect an important mechanical structure—such as chemotherapy drugs depolymerizing the cytoskeleton [12]—or a mechanosensor, cell stiffness is an excellent test of such damage. Likewise, if the disease in question affects the mechanics [2,5], rheology can indeed be used to measure efficacy.

### 4.3. Application to Drug Effects

The only study so far that explicitly investigates drug efficacy using viscoelastics was published in 2023 by Ma et al. (Figure 15) [63]. In this work, the authors perform force relaxation measurements once an hour for 8 h and fitted the data to a five-element Maxwell model (Figure 12f), hence ending up with five fitting parameters. As a sixth parameter, they also measured Young’s modulus using the approach curve.

Treating human breast cancer cells (MCF-7) with the chemotherapeutic drugs paclitaxel (PTX) and doxorubicin (DOX), they found that they were indeed able to measure increased stiffness and viscosity whereas the apparent Young’s modulus showed less change compared to the uncertainty. Also in 2023, Weber et al. investigated drug treatment of cancer cells [105].

## 5. Additional and Alternative Approaches

The following approaches are expansions upon or alternatives to the techniques already presented. Researchers more established in the area might benefit from considering incorporating these advanced approaches in their work.

### 5.1. High-Throughput Techniques

Due to the slow speed of AFM, a major challenge is measuring enough cells to gain statistically significant results. However, recent studies have shown AFM techniques where an extraordinarily large number of cells could be measured [104,106] and which may be of interest to the reader. Most noticeably, a 2020 study used machine learning to automatically measure over 1000 cells either with or without drug treatment [106]. Each sample could be measured over 2 h with no significant change due to a live incubator with temperature and CO_2_ control. The authors clearly show the effect of caspofungin on yeast cells (Figure 16), but they also point out that a smaller experiment could have come to contradictory results due to the high variability and bimodal distribution of the cell population.

### 5.2. Tomography

Throughout this review, the focus has been on fitting a single material model to the force–distance or force–time curves. However, it is reasonable to think that the mechanics of deeper-lying structures will affect the measurement of greater indentation depths. Therefore, some researchers perform AFM tomography to arrive at a 3D view (Figure 17) of the cell’s mechanical state [107,108]. This could potentially be valuable to drug testing, as the mechanical change can thereby be localized to a specific organelle.

While any tip can be used for tomography, some groups grow ultrathin nanoneedles which not only deform but delicately penetrate the different parts of the cell while the force is monitored. This type of tomography is referred to as nano-endoscopy [109].

### 5.3. Standardization Efforts

Another approach to improving reproducibility is to define a common standard for how cells are measured [110,111,112,113]. To achieve this goal, the Standardized Nanomechanical AFM Procedure (SNAP) was introduced in 2017 [111] and it has since been built into the MATLAB program *AFMech Suit* [110].

### 5.4. Non-AFM Options

While this review is exclusively focused on the use of AFM, other cell mechanics measurement methods [114] (Figure 18) might be of interest to the reader and which in some cases have already been used to observe drug effects. The theoretical understanding of cell mechanics is still directly transferable.

For instance, micropipette aspiration measures whole-cell stiffness by sucking a section of a cell into a micropipette. By controlling the pressure and observing the cellular deformation with an optical microscope, cell stiffness can be calculated [115,116,117].

A strong competitor to AFM is particle-tracking microrheology [118] in which particles are tracked inside the cell (Figure 18G) which the AFM tip normally does not reach. In this way, it is possible to gain information about their diffusion, including the amplitude of movements and directionality [2].

Scanning Ion Conductance Microscopy is similar to AFM, but it images the cell with a pipette measuring ion conductance without directly touching the cell [93]. This does result in a lower spatial resolution, but it has proven capable of measuring cell membrane fluctuations [81] which can be related to the effect of drugs [119].

A more macroscopic alternative is using a parallel plate rheometer, some of which can measure cell monolayers to observe the total rheology of an entire cell culture [120].

## 6. Outlook

It is likely that AFM—if it stands the test of time—will be commercially developed to be as easy and effective to use as standard instruments used in biomedical research as well as the pharmaceutical industry today. However, given the popularity of non-AFM methods for cell mechanics, it is entirely possible that other methods—especially those that are already fast—will come to dominate the field. The need for easier and faster measurements has also been pointed out earlier [42,69].

This is partly due to the major lack of good software solutions to easily analyze force data, similar to how Gwyddion has become universal to analyzing topographical AFM images. The potential for publishing new software solutions here is large, especially ones incorporating machine learning (such as *Nanite* [53]) which might address the limitations of traditional modeling.

A related issue is that the broader field of cell mechanics is itself under development, making it difficult to interpret experimental results. The sooner a fundamental understanding of cell mechanics comes out, the sooner the results of drug tests can be meaningfully understood. It might, therefore, be fruitful for researchers to collaborate closely with biophysicists.

Since mostly basic elasticity has been studied (Figure 5d), there is ample room for studies connecting drug effects with non-ideal deformations (Figure 3) and rheology (Figure 10). It should also be noted that while there are studies on efficacy (e.g., [63]), studies directly addressing potency appear to be missing, likely because of the added challenge of following the cells over time.

From the perspective of the theory of science [121], it has been proposed that the development of a scientific field can be divided into four phases (Figure 19) [122]:

In the first phase, the field is established and questions are asked. In the second phase, the methods are developed. The third phase consists of an explosive development with many discoveries due to the foundation laid by the first two phases while, in the fourth phase, standards are developed and textbooks are written, thus solidifying the field.

Given the current progress of cell mechanics in pharmaceutics, the field is currently in phase two. This field has been established with questions such as measuring the efficacy and toxicity of drugs, and the first results have also come in. However, the methods come from outside of pharmaceutical research, and findings are likewise published in non-pharmaceutical journals. (Figure 5a,b). Once the methods are properly developed for this field—also in terms of hardware and analysis software—the scientific field will move into the third phase and cell mechanics will likely have very fertile applications in pharmaceutics in the near future.

## 7. Conclusions

An overview of cell mechanics in pharmaceutics using AFM has been presented. AFM is showing promising results with its ability to track the uptake of a drug over time as well as being able to show the efficacy of a drug, especially in terms of cytotoxicity. In particular, rheology has shown newer and clearer results on efficacy. Researchers have also already combined AFM with other techniques such as fluorescence microscopy and Western blotting to gain even more information.

While the scientific field is not yet the right choice to achieve fast and clear information on a drug or delivery system, it is perfect for researchers looking to participate in a newly developing research area by developing its methodologies. In particular, there is great potential in interdisciplinary collaborations with biophysicists and others specializing in cell mechanics.

## Figures and Tables

**Figure 1 pharmaceutics-16-00733-f001:**
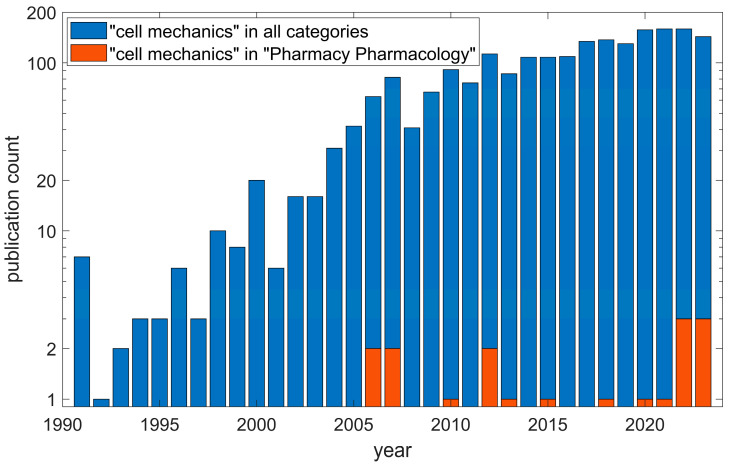
Research trend analysis of cell mechanics up to and including the year 2023. Note the logarithmic y-axis. The data are from Clarivate Web of Science. © Copyright Clarivate 2024. All rights reserved.

**Figure 2 pharmaceutics-16-00733-f002:**
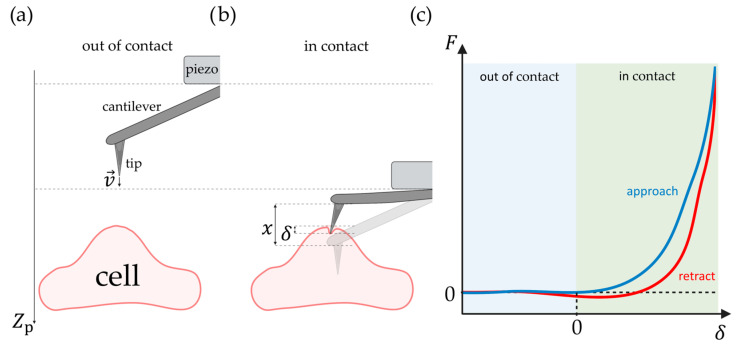
Illustration of nanoindentation by AFM. (**a**) The piezo moves the tip toward the cell at a speed $\vec{v}$ while it is still out of contact. (**b**) The tip indents a distance δ into the cell. The cantilever is correspondingly bent at a distance x. The piezo has moved a distance $Z_{p}$. (**c**) Generic sketch of a force–indentation curve with the approach (blue) and the subsequent retraction (red). This illustration was created using BioRender.com.

**Figure 3 pharmaceutics-16-00733-f003:**
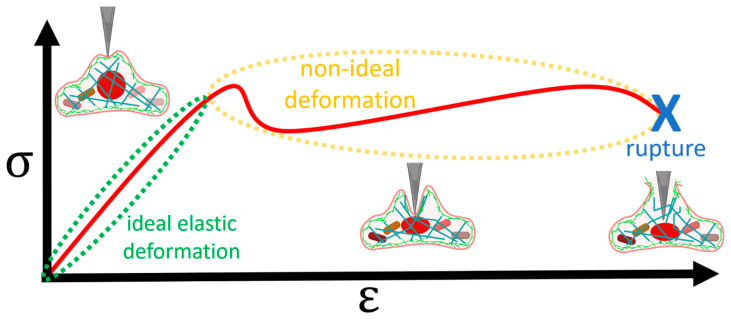
Cells are ideally elastic (green) when stress and strain are proportional, which is the case at small deformations of the cell cortex. However, past a certain load, the cell’s inhomogeneity causes deviations from this ideality (yellow) [30]. At a sufficiently high strain, the cell ruptures completely (blue). The red curve is a generic stress–strain curve based on [28]. The cell illustrations were created with BioRender.com.

**Figure 4 pharmaceutics-16-00733-f004:**
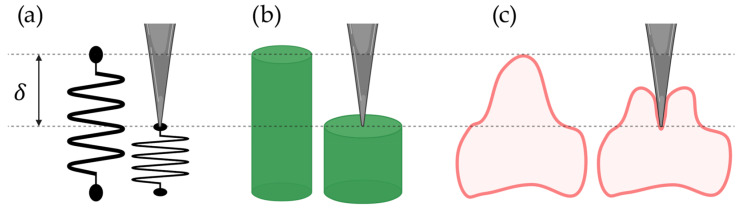
Illustration of the different views of indentation at a depth δ. (**a**) Simple compression of a spring. (**b**) Even compression of a cylinder with a positive Poisson ratio. (**c**) Realistic uneven compression of a cell. This illustration was created using BioRender.com.

**Figure 5 pharmaceutics-16-00733-f005:**
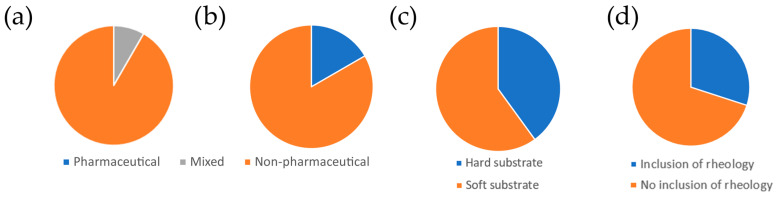
Meta-analysis on the reviewed experimental papers. (**a**) Fractions of studies performed by similar pharmaceutical institutes. (**b**) Studies published in pharmaceutical vs. non-pharmaceutical journals. (**c**) Fraction of cells seeded on a hard substrate (glass) vs. a soft substrate. (**d**) Fraction of studies including rheological measurements.

**Figure 6 pharmaceutics-16-00733-f006:**
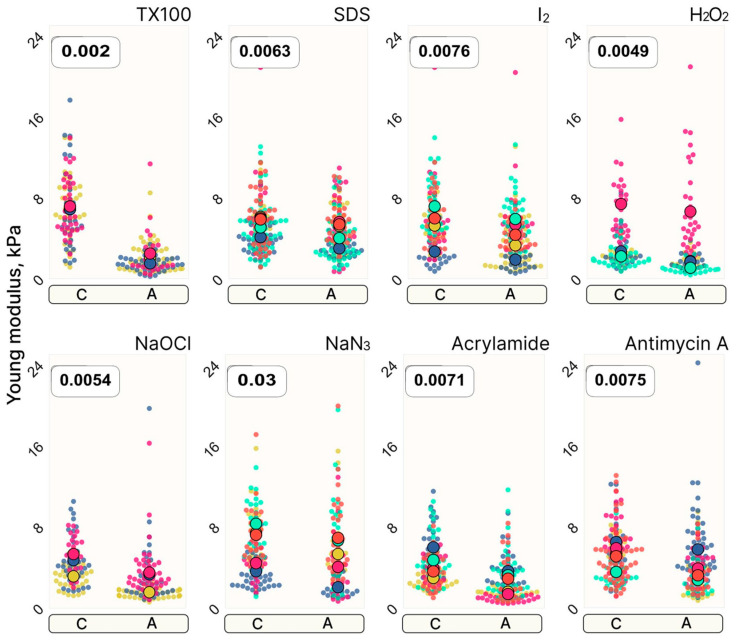
Stiffness changes in cells treated with one of eight different chemicals. While an effect could be measured, the relative variability was also large. Large dots represent group means while small dots represent each individual measured cell. Control (C) and treated (A) groups measured on the same day have the same color. Reproduced with permission from [69].

**Figure 7 pharmaceutics-16-00733-f007:**
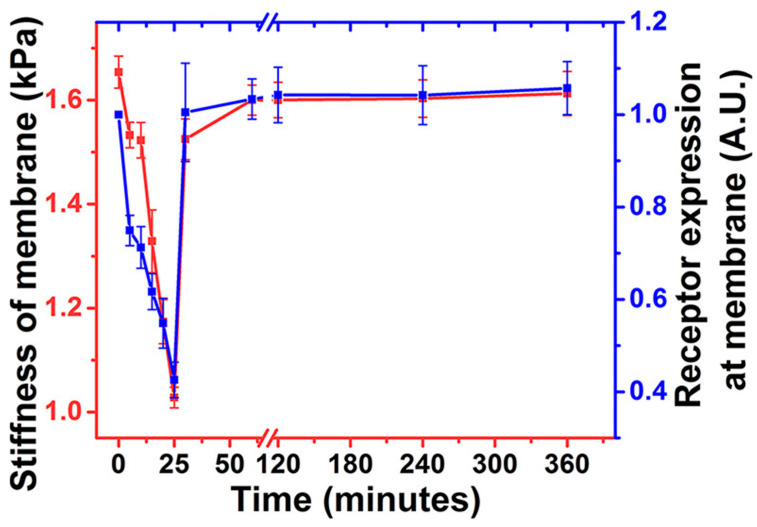
Using stiffness to track the uptake of gold nanoparticles. The stiffness change could be correlated with receptor expression. Adapted with permission from [15]. Copyright 2024 American Chemical Society.

**Figure 8 pharmaceutics-16-00733-f008:**
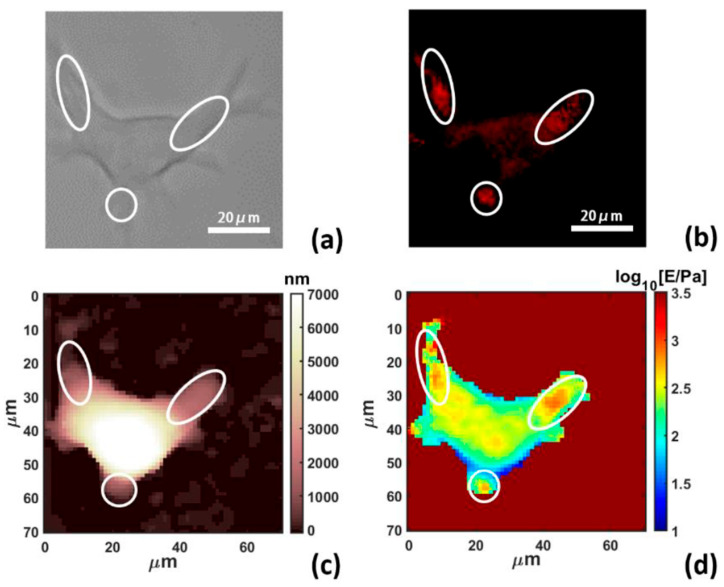
Combined force volume mapping and fluorescence microscopy show a correlation between nanofibers and increased stiffness. (**a**) Differential Interference Contrast image. (**b**) Fluorescence image showing the position of the nanofibers. (**c**) Topographical map. (**d**) Stiffness map with stiffness peaks in the same location as the nanofibers. Reproduced from [59] under the Creative Commons license CC BY 4.0 DEED.

**Figure 9 pharmaceutics-16-00733-f009:**
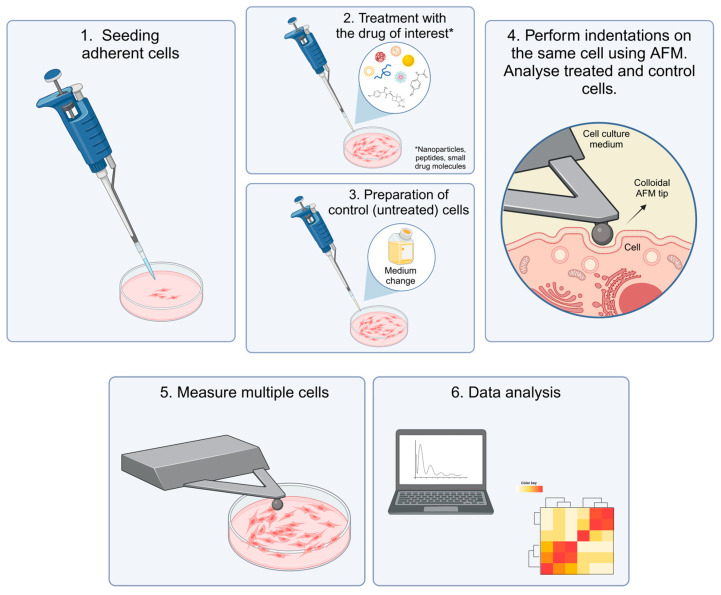
Recommended procedure for achieving significant cell mechanics results with AFM. This graphic was created using BioRender.com.

**Figure 10 pharmaceutics-16-00733-f010:**
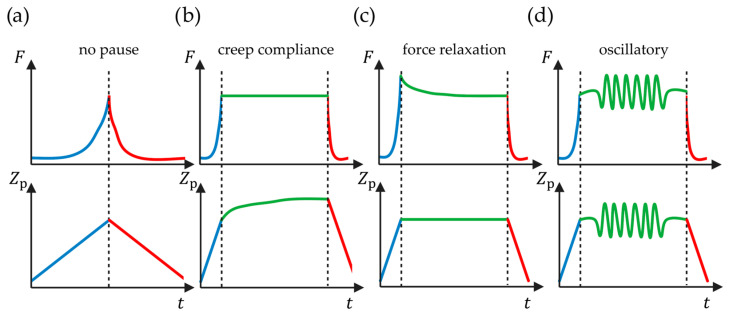
Time-dependent measurements for cell rheology approach (blue), pause (green), and retract (red). In the top row are the force–time curves, while the bottom row shows the piezo positions over time. (**a**) The force-time representation of a regular force-distance measurement (as in Figure 2c). (**b**) Creep compliance where the force is kept constant. (**c**) Stress relaxation where the piezo position is constant. (**d**) An oscillatory measurement where the piezo oscillates and the force therefore also oscillates. The illustration is based on [2,43] and was created using BioRender.com.

**Figure 11 pharmaceutics-16-00733-f011:**
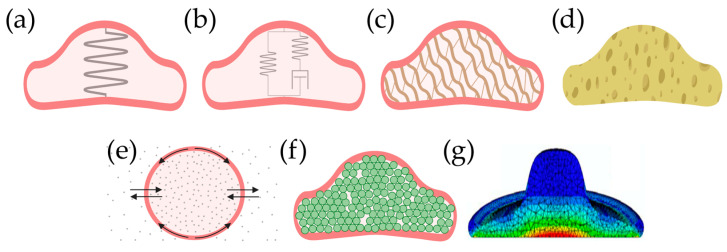
Qualitative illustrations of the select mechanical models for cells. (**a**) Ideal elastic. (**b**) Spring-dashpot. (**c**) Polymer network. (**d**) Poroelastic. (**e**) Semi-permeable water balloon. (**f**) Soft-glassy. (**g**) Finite element analysis where the colors represent strain values. (**a**–**f**) were created using BioRender.com, while (**f**) was adapted from [92] under the Creative Commons license CC BY 3.0 Deed.

**Figure 12 pharmaceutics-16-00733-f012:**
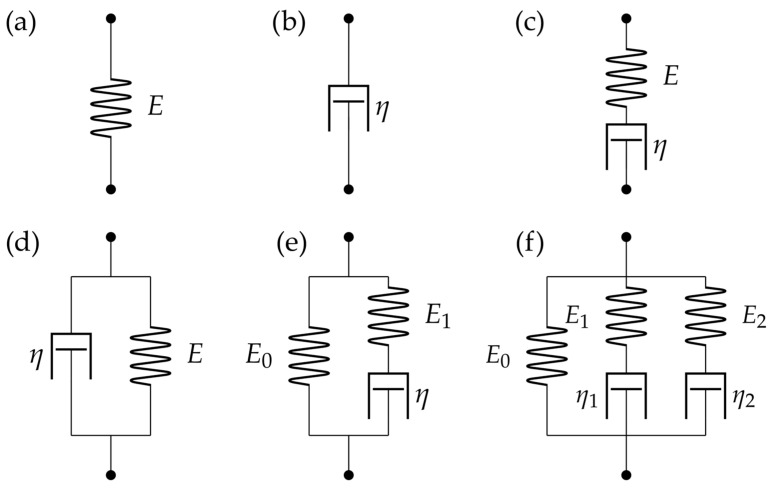
(**a**) A spring represents elasticity in the model as described by Hooke’s law (Equation (6)). (**b**) A dashpot represents viscosity as described by Equation (29). The two can be combined (**c**) in series as the Maxwell model or (**d**) in parallel as the Kelvin-Voight model. Larger combinations include (**e**) Standard Linear Solid (SLS) and (**f**) the five-element Maxwell model. Illustration based on [29].

**Figure 13 pharmaceutics-16-00733-f013:**
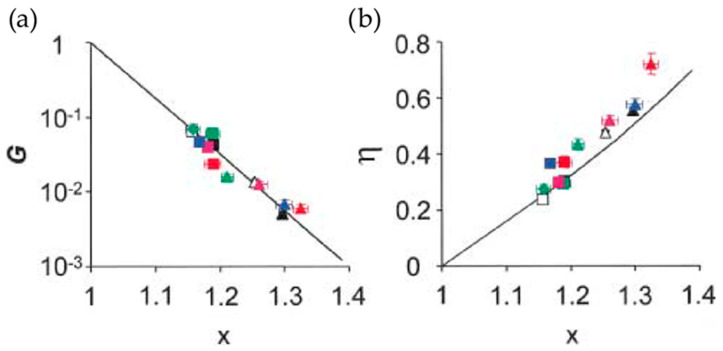
Master curves reported by Fabry et al. for (**a**) normalized stiffness G and (**b**) the loss tangent η as a function of x. Filled-in squares (■) represent control cells while the other data point shapes represent treatmeant with different drugs and yet all lie on the same line. Each color represents a distinct cell line. Adapted with permission from [97].

**Figure 14 pharmaceutics-16-00733-f014:**
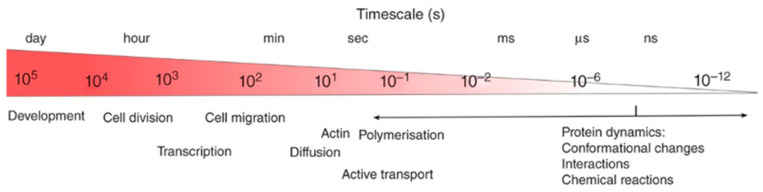
Knowledge of the timescales for the various processes can be used to interpret results. Adapted with modifications with permission from [2].

**Figure 15 pharmaceutics-16-00733-f015:**
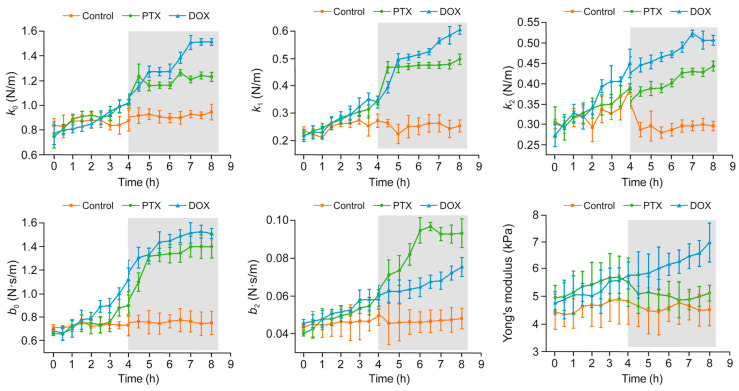
Plot of the viscoelastic parameters over time for cells treated with paclitaxel (PTX) and doxorubicin (DOX). Here, $b_{0}$ and $b_{2}$ refer to the viscosities of the two Maxwell elements. A clear difference from the control past the 4 h mark can be observed in all parameters, but less so for Young’s modulus. Adapted with permission from [63].

**Figure 16 pharmaceutics-16-00733-f016:**
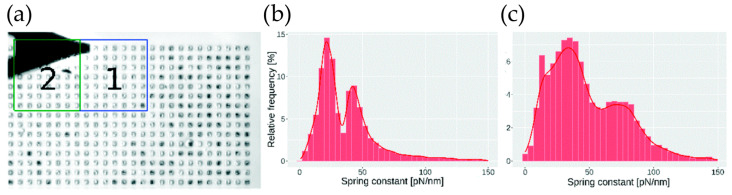
Automatic measurement of thousands of cells before and after treatment with caspofungin. (**a**) The multi-cell experimental setup. (**b**) Cell stiffness before treatment. (**c**) Cell stiffness after treatment. Note that the authors in this experiment describe the cell stiffness with a spring constant (Equation (5)). Adapted with permission from [106].

**Figure 17 pharmaceutics-16-00733-f017:**
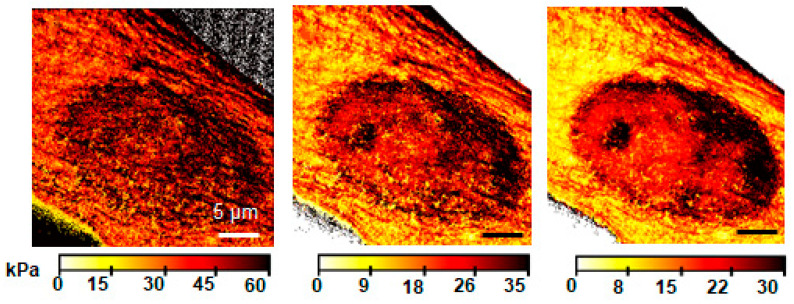
Mapping Young’s modulus, the cell nucleus becomes more pronounced for larger indentation depths: (**left**) 50–200 nm, (**middle**) 200–400 nm, and (**right**) 400–850 nm. The graphic was provided by Ricardo Garcia with data from [108].

**Figure 18 pharmaceutics-16-00733-f018:**
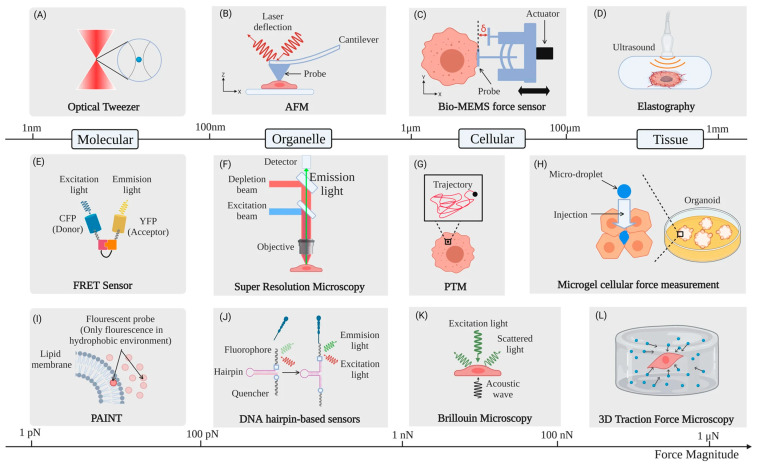
Selection of different experimental techniques for cell mechanics and imaging along with typical forces and length scales. (**A**) Optical tweezers, (**B**) atomic force microscopy, (**C**) bio-MEMS, (**D**) elastography, (**E**) fluorescence resonance energy transfer (FRET), (**F**) super resolution microscopy, (**G**) particle tracking micro-rheology (PTM), (**H**) microgel cellular force measurement, (**I**) PAINT, (**J**) DNA hairpin-based sensors, (**K**) Brillouin microscopy, and (**L**) traction force microscopy. Reproduced from [8].

**Figure 19 pharmaceutics-16-00733-f019:**
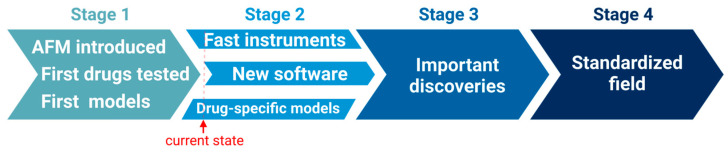
In the model by Shneider [122], AFM for cell mechanics in pharmaceutics is now in stage two of its development as a scientific field. Created using BioRender.com.

## Data Availability

Data are contained within the article and Appendix A.

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
