# Peer review of "Atomic Force Microscopy for the Study of Cell Mechanics in Pharmaceutics"

_pharmaceutics, 2024, doi:10.3390/pharmaceutics16060733_

Round 1

Reviewer 1 Report

Comments and Suggestions for Authors

This paper deals with the review of cell surface mechanism with pharmaceutics by atomic force microscopy.  The theme is interesting, so that the paper can be published in this journal.  If the following points were added, the paper would be more attractive:

    Shape of AFM cantilever tip is important for observing roughly surface.  The authors would mention the shape simply.

    Usually animal cells form a mass: single cell is separated from it.  In the case, the observers pre-treated the cells.  The authors would mention the importance of pre-treatment.  Residual intercellular substances would be also mentioned for influencing the measurements.

Author Response

We have added a point-to-point list with a Word document, please see the attachment:

Reviewer 2 Report

Comments and Suggestions for Authors

The authors are right in writting about this subject in the sense that cell mechanics is increasing its importance due to previous results in biology studying celular and pathological processes.

- The review summarizes the basic issues of the field and describes the numerous difficulties that a non-expert researcher may currently encounter. However, the authors practically do not comment on important things such as cell immobilization, operating conditions such as air or fluid or other atmospheric parameters, temperature and AFM probes. These are things that greatly influence these experiments and that must be discussed.

- I'm also missing an actual conclusions section outlining the main clear conclusions in the field to date.

- Once you described atomic force microscopy as AFM, you should continue citing this technique as AFM and not in the two ways

- In line 54 you say that tapping mode is an intermittent contact mode, but this is not correct. Tapping mode belongs to the dynamic operational modes.

- In 2.1 you say that to measure stiffness, the tip approaches to the cell and …? But you should explain that first a loading force should be set –very important-, then the tip approaches to a cell and if this applied force is in the range in which it is sufficient to introduce a deformation in the membrane but without breaking it, an indentation curve will be made, and therefore the mechanical response to that applied force can be studied. Not all the force-distance curves are indentation curves, and this should be clear.

- In line 84 you say that the tip can be functionalized with single molecules but this is not possible, although to study intra- and inter-molecular forces, tips are functionalized with a set of molecules and then what is determined is the unbinding force of a single molecule or of a single molecular complex. 

- In line 222, in “3.1.1. Cytotoxicity assay”, I need a more in-depth discussion based on published measurements about how cytotoxicity can be related with stiffness, or other similar nanomechanical parameters.

- In line 647, in “ 5.1. High-throughput techniques”, I do not know what type of approaches are you referring to. You have to describe them better.

- There are references with different wrong formats; in some the doi or other data is missing

- There are numerous errata, phrases of dubious meaning, misplaced singulars and plurals, spaces, commas incorrect character sizes, and periods or inapropiate italicss in places as Figure 13 caption. The numerous surprising “Error! Reference source not found” errors are incredible.

Comments on the Quality of English Language

Minor errors

Author Response

(The authors gave the same response as above.)

Round 2

Reviewer 2 Report

Comments and Suggestions for Authors

I consider that the changes introduced in this latest version have significantly improved the review, making it much more complete and describing the main aspects involved in this type of measurements, since they are complex.

As comments to improve:

- I think that the quality of the figures in general can be improved.

- In 2.2. Contact mechanics, you say “To quantify cell mechanics…”, and this should be “To analyze cell mechanics” or “To estimate cell mechanics parameters”.

- In 2.3. Choice of AFM probe, you say “…where  will lie in the range of  …“, and you should say 0.60, and not 0-6, nN/nm or N/m.

Author Response

Thank you for your comments, we have added a file with our point-to-point response.
